# Comprehensive Analyses of the Histone Deacetylases Tuin (HDT) Gene Family in Brassicaceae Reveals Their Roles in Stress Response

**DOI:** 10.3390/ijms24010525

**Published:** 2022-12-28

**Authors:** Pan Xie, Wei Liu, Rui Ren, Yu Kang, Yan Liu, Yuan Jia, Lunwen Qian, Xin He, Chunyun Guan

**Affiliations:** 1Southern Regional Collaborative Innovation Center for Grain and Oil Crops in China, Hunan Agricultural University, Changsha 410128, China; 2College of Landscape Architecture and Art Design, Hunan Agricultural University, Changsha 410128, China; 3Oil Crops Research Institute, Hunan Agricultural University, Changsha 410128, China; 4Hunan Branch of National Oilseed Crops Improvement Center, Changsha 410128, China

**Keywords:** *Brassica napus*, *HDT*, abiotic stresses, freezing, hormone

## Abstract

*Histone deacetylases tuin* (*HDT*) is a plant-specific protein subfamily of histone deacetylation enzymes (*HDAC*) which has a variety of functions in plant development, hormone signaling and stress response. Although the *HDT* family’s genes have been studied in many plant species, they have not been characterized in *Brassicaceae*. In this study, 14, 8 and 10 *HDT* genes were identified in *Brassica napus*, *Brassica rapa* and *Brassica oleracea*, respectively. According to phylogenetic analysis, the *HDT*s were divided into four groups: *HDT1*(*HD2A*), *HDT2*(*HD2B*), *HDT3*(*HD2C*) and *HDT4*(*HD2D*). There was an expansion of *HDT2* orthologous genes in *Brassicaceae*. Most of the *HDT* genes were intron-rich and conserved in gene structure, and they coded for proteins with a nucleoplasmin-like (NPL) domain. Expression analysis showed that *B. napus*, *B. rapa*, and *B. oleracea HDT* genes were expressed in different organs at different developmental stages, while different *HDT* subgroups were specifically expressed in specific organs and tissues. Interestingly, most of the *Bna/Br/BoHDT2* members were expressed in flowers, buds and siliques, suggesting they have an important role in the development of reproductive organs in *Brassicaceae*. Expression of *BnaHDT* was induced by various hormones, such as ABA and ethylene treatment, and some subgroups of genes were responsive to heat treatment. The expression of most *HDT* members was strongly induced by cold stress and freezing stress after non-cold acclimation, while it was slightly induced after cold acclimation. In this study, the *HDT* gene family of *Brassicaceae* was analyzed for the first time, which helps in understanding the function of *BnaHDT* in regulating plant responses to abiotic stresses, especially freezing stresses.

## 1. Introduction

Posttranslational modifications of histones in plant genomes, such as acetylation, methylation, phosphorylation and ubiquitination, establish a rapid and reversible pattern of gene expression. The steady-state acetylation level of histones is achieved through the action of histone acetyl transferases (HATs) and histone deacetylation enzymes (*HDACs*) by adding an acetyl group to the N-terminal of histones, which is a marker of transcriptional activation in eukaryotes. Acetylation involves the acetylation of two amino acids, lysine and arginine. Compared with arginine, lysine acetylation is more conservative and more thoroughly studied [1]. Histone acetylation has been reported in different lysine residues of histones, such as H3 (K4, K9, K14, K18, K23 and K27), H4 (K5, K8, K12 and K16) and H2B (K5, K12, K15 and K20) [2]. Acetylation of lysine residues at the tail of histones reduces the amount of positive charge carried by histones and reduces their affinity to negatively charged DNA strands, which results in the unwinding of local DNA and histone octamers, thus promoting the binding of various protein factors involved in transcriptional regulation with DNA-specific sequences and then playing a transcriptional regulation role [3,4].

The *histone deacetylases* (*HDAC*) are a supergene family that is widespread in eukaryotes, including yeast, mammals and plants. *HDAC*s are divided into three families: the *RPD3/HDA1* family, *SIR2* family and *HDT* family. The *HDT* (*HD2*) family is plant-specific *histone deacetylase* [5], which is completely different from the *RPD3/HDA1* family in sequence [6]. It has also been shown that *HDA* evolution is associated with increased structural ductility/disorder, and two *Brassicaceae*-specific *HDAs* have been identified, as well as key mutations that affect the catalytic activity of individual *HDA*s [7]. RPD3 and SIR2 share sequence homology with yeast *HDACs*, whereas the *HDT* family shares no sequence homology with yeast *HDACs* [8,9].

More and more data have shown that *HDACs* play a key role in plant growth and development, including bud, flower [2,10,11], seed [3,12] and root development [13], and that they respond to biotic and abiotic stresses, such as drought [14], salt stress [15], low temperature [16], and ABA [5,17,18]. In *Arabidopsis*, there are four *AtHDT* members, AtHDT1 (*AtHD2A*), *AtHDT2* (*AtHD2B*) [12], *AtHDT3* (*AtHD2C*) [19] and *AtHDT4* (*AtHD2D*) [20,21]. *AtHDT1*, *AtHDT2* and *AtHDT3* can mediate transcriptional inhibition in *Arabidopsis* [14,22]. *AtHDT1* is highly expressed in *Arabidopsis* flowers and young siliques, while *AtHDT2* is widely expressed in stems, leaves, flowers and young siliques [14]. Overexpression of *AtHDT1* in *Arabidopsis* results in leaf abnormalities, delayed flowering and seed abortion [19]. *Arabidopsis HDT1* and *HDT2* affect leaf morphological development by regulating the expression of miR165/166 in *Arabidopsis* [23]. *AtHDT3* is inhibited by abscisic acid, and its overexpression confers ABA sensitivity phenotypes, reduces transpiration and enhances tolerance to salt and drought stress [18].

*HDTs* have been studied in crops such as rice, soybean, tomato and potato [24,25]. Overexpression of *HDT701* induced early flowering of hybrid rice under long-day conditions compared with the parents. *HDT702* knockdown resulted in abnormal plant height and narrow leaf and stem segments [26]. *ScHDT1* in potato (*Solanum chacoense*) is a homolog of *Arabidopsis HDT1*, which has been shown to increase the accumulation of transcripts in ovules after fertilization [27]. *SlHDT3* was a positive regulator of fruit ripening by affecting ethylene synthesis and carotenoid accumulation [15,25]. 

*HvHDAC2-1* in barley was induced by jasmonic acid (JA), ABA and salicylic acid (SA) [28]. Studies show that two *HDTs* act as key negative regulators of elicitor-induced cell death in tobacco [29]. Overexpression of *HDT701* in rice decreases ABA, salt and osmotic stress resistance during seed germination [24]. In *Arabidopsis*, High Expression of Osmotically Responsive Gene (HOS15) interacts with *HDT3* in the promoter region of the *Cold-responsive* (*COR*) gene, thereby mediating *HDT3* degradation, leading to upregulation of the *COR* gene as part of the cold stress response [30]. 

*B. napus* (2n = 38, AACC) is a heterotetraploid hybrid species formed by the natural hybridization of *Brassica rapa* (2n = 20, AA) and *Brassica oleracea* (2n = 18, CC) through natural chromosome doubling [31]. In this study, we investigate the significant role of *HDT* genes in *Brassicaceae* plants. Therefore, we identified 32 *HDT* genes of *B. napus*, *B. rapa* and *B. oleracea* and compared their gene structures, chromosomal locations, evolutionary relationships and expression patterns in different tissues and under different abiotic/biotic stresses and plant hormonal treatments. Thus, the comprehensive analysis in this study provides a fundamental understanding of *HDAC*s in development and stress responses in *Brassicaceae* plants. 

## 2. Results

### 2.1. Identification and Characterization of HDT Genes in B. napus, B. rapa and B. oleracea

To identify HDT proteins in *B. napus*, *B. rapa* and *B. oleracea*, we performed a BLASTp search against the annotated proteins of *B. napus*, *B.rapa* and *B. oleracea* in Ensembl Plants (http://plants.ensembl.org/index.html, accessed on 13 August 2021) using *Arabidopsis AtHDT* protein (*AtHDT1*, *AtHDT2*, *AtHDT3* and *AtHDT4*) sequences as queries. Sets of 14 (1 *BnaHDT1*, 10 *BnaHDT2*, 1 *BnaHDT3* and 2 *BnaHDT4*), 8 (6 *brHDT2*, 1 *brHDT3* and 1 *brHDT4*) and 10 (1 *BoHDT1*, 7 *BoHDT2*, 1 *BoHDT3* and 1 *BoHDT4*) HDT proteins were identified in *B. napus*, *B. rapa* and *B. oleracea*, respectively (Appendix A).

All 32 *HDT*s in *B. napus*, *B. rapa* and *B. oleracea* contained conserved NPL domains. Among them, 27 HDT proteins were predicted to localize in the nucleus by using ProtComp Version 9.0; these protein are acidic proteins with PI values of less than 5, while 5 HDT2 proteins (*Bra020207, Bo2g024200, Bo2g024210, Bo2g024220* and *BnaA02g05820D*) were weakly acidic or alkaline proteins, having no firm predicted subcellular localization, and their PI values were all greater than 6 (Table 1 and Appendix A).

All the *HDT* genes are divided into typical *HDT* and short abnormal *HDT* categories. Typical *HDT* gene members are *HDT1*, *HDT2*, *HDT3* and *HDT4*, which have 215–356 amino acids and 5–11 exons (Table 1). It was discovered using isoelectric dot analysis that *HDT* is a typical acidic protein with a typical acidic region. Additionally, there are six short abnormal genes (*Bra020206*, *Bo2g024150*, *Bra020204*, *BnaA02g05810D*, *BnaA02g05790D* and *BnaCnng34610D*), which typically contain 162–184 amino acids and have three exons and two introns. According to gene structural analysis, all are lacking the last few amino acids of the carboxy-terminus. There is also a long *HDT* gene (*Bo2g024200*). The first half of the long gene is the typical structure of *HDT*, and the second half results from a fusion with another sequence (Appendix A).

### 2.2. Phylogenetic Analysis and Chromosomal Locations of HDT Genes in B. napus, B. rapa and B. oleracea

To explore the classification and evolutionary characteristics of the HDT proteins, an unrooted phylogenetic tree based on the 36 protein sequences of *B. napus* (14), *B. rapa* (8), *B. oleracea* (10) and *Arabidopsis* (4) HDT genes was constructed in MEGA X (Figure 1). According to the phylogenetic analysis, the *HDT* genes were divided into four groups: *HDT1* (homologous to *AT3G44750.1*/*AtHDT1*), *HDT2* (homologous to *AT5G22650.1*/*AtHDT2*), *HDT3* (homologous to *AT5G03740.1*/*AtHDT3*) and *HDT4* (homologous to *AT2G27840.1*/*AtHDT4*). *HDT2* has six sets of homologs, while the other three have only one each. These results indicated that there was an expansion of *HDT2* homologous genes in *B. napus*, *B.rapa* and *B. oleracea*. 

A neighbor-joining phylogenetic tree was generated by MEGA X with full-length HDT sequences (1000 bootstrap replicates). The resulting four groups classified into four resulting subfamilies (*HDT1*, *HDT2*, *HDT3* and *HDT4*, highlighted in purple, pink, red and blue, respectively) are labeled. According to the homologous gene sets among the Ar (*B. rapa*,), Co (*B. oleracea*), An and Cn subgenomes of *B. napus*, six Ar-Co-An-Cn pairs were identified among the 32 *Bna*/*Br*/*BoHDT* genes (Figure 1). The A genome of *B. napus* lacks the homologous genes corresponding to the *B. rapa HDT1* gene *Bra009513*, and the C genome lacks the homologous genes corresponding to the *B. oleracea HDT2* genes (*Bo2g024210*, *Bo2g024220* and *Bo2g024200*), while the homologous genes in the *HDT3* and *HDT4* subgroups were relatively conserved (Figure 2 and Appendix A).

The An and Cn subgenomes of *B. napus* were collinear with the corresponding diploid Ar and Co genomes, most of the An-Ar and Cn-Co homologous pairs showed similar chromosomal locations [32]. As shown in Figure 2, the 14 *BnaHDT* genes were unevenly distributed on the 10 chromosomes of *B. napus* (chrA02-A04, A10, C01-C04, chrC09 and chrCnn_random. There was one tandem gene, *BnaA02g05790D*/*BnaA02g05810D*/*BnaA02g05820D*/*BnaA02g05840D*, on chromosome A02 in *B. napus*, which was homologous to the tandem gene *Bra020204*/*Bra020206*/*Bra020207*/*Bra020209* on chromosome A02 in *B. rapa*, while the tandem pair (*Bo2g024150*/*Bo2g024200*/*Bo2g024210*/*Bo2g024220*/*Bo2g024240*) corresponding to *B. olearcea* on chromosome C02 in *B. napus* was lost (Figure 2 and Appendix A). Therefore, it was inferred that the tandem genome was mutated in the homologous evolution process.

### 2.3. Motif Analysis (MEME) of HDT

To further analyze the domains and motifs of the HDT proteins, 10 motifs were predicted by MEME (Appendix A) (http://meme-suite.org/tools/meme, accessed on 24 May 2022). As shown in Figure 3B, all HDT proteins contain Motif 3, and Motif 2-1-5-3 constitutes the NPL domain (Appendix A). All of the HDT1 proteins contain Motif 2-5-3-8-6-7-4-9. HDT2 proteins contain Motif 2-1-3-7, except *Bo2g024210*, which only contains Motif 2-1-3. HDT3 proteins contain Motif 2-1-5-3-6-6-7-4-9. HDT4 proteins contain Motif 2-5-3-6-7, while *AT2G27840* only contains Motif 3-6-7. Two of the six short abnormal genes, Bra020206 and *BnaA02g05810D*, lack Motif 4-6-10, while *Bo2g024150*, *Bra020204*, *BnaA02g05790D* and *BnaCnng34610D* lack Motif 4-9-10. The long *HDT* gene (*Bo2g024200*) contains Motif 2-1-5-3-8-2-1-5-3-7 (Figure 3B).

### 2.4. Expression Profiling of HDT Genes in Different Tissues

Based on *Arabidopsis* eFP Browser data (http://bar.utoronto.ca/efp/cgi-bin/efpWeb.cgi, accessed on 24 May 2022) and RNA-Seq data (*B. rapa*: GSE43245, B. oleareaca: GSE42891 and *B. napus*: PRJNA394926) (Appendix A) [33,34,35], the *HDT* gene was expressed in different vegetative and reproductive organs of the four species at different developmental stages (Figure 4). In general, the expression pattern of *HDT* differed between groups, but the expression pattern of the four species within the same subgroup was very similar. In *Arabidopsis*, *AtHDTs* were highly expressed in roots, carpels and flowers; *BoHDTs* and *BrHDTs* were highly expressed in leaf, bud and silique tissues in *B. rapa* and *B. olearcea*. In *B. napus*, *BnaHDTs* were highly expressed in bud, stamen and ovule tissues, as well as peel and silique tissues. The expressions of *AtHDT2* and corresponding genes in *B. rapa*, *B. olearcea* (*Bo2g024240* and *Bra020209*) and *B. napus* (*BnaC02g09720D* and *BnaA02g05840D*) were higher than those of other *HDT* genes in all tissues. In *HDT2*, the expression patterns of a pair of genes (*Bo3g016960* and *Bra006629*) in *B. olearcea* and *B. rapa* were highly similar with corresponding pairs of gene (*BnaC03g10890D* and *BnaA03g08590D*) in *B. napus*, and the expression levels in stem, bud and silique tissues were higher than those in other tissues. Another pair of genes (*Bo2g024150* and *Bra020204*) in *B. olearcea* and *B. rapa* had an expression pattern highly similar to that of a corresponding pair of genes (*BnaCnng34610D* and *BnaA02g05790D*) in *B. napus*. These genes are specifically expressed in bud, stamen and silique tissues. All the results suggested that *Bna/Br/BoHDT2* members showed expression in flowers and siliques, which indicates their importance in propagative organ development in *Brassicaceae* plants.

### 2.5. Expression Profiling of HDT Genes under Abiotic Stress and Phytohormone Treatments

To reveal the roles of *BnaHDTs* in stress responsiveness in *B. napus*, their expression patterns upon various abiotic and phytohormone treatments were investigated (Appendix A). As shown in Figure 5, the expression levels of most genes were increased after treatment with ABA for 6 h and ethylene treatment for 6 h. The *HDT2* and *HDT3* subgroups were more responsive to heat shock than the other two subgroups. Three genes in *HDT2* (*BnaA02g05840D*, *BnaC02g09720D* and *BnaA10g13800D*) and one gene in *HDT3* (*BnaCnng02600D*) were increased sharply after 12 h of heat shock. In addition, the expression of *BnaA02g05840D* in *HDT2* was strongly induced after 12 h of low-temperature treatment. Overall, the *HDT* gene family is responsive to multiple hormonal and adversity stresses.

### 2.6. Expression Profiling of HDT Genes under Low Temperature Stress

As mentioned previously, the two genes of *HDT2* were highly expressed in all tissues and responded strongly to cold stress (Figure 5). In order to elucidate the potential function of *BnaHDT* in response to low-temperature stress, the transcriptional patterns of *BnaHDT* in *B. napus* were studied after different low-temperature stresses under conditions of cold acclimation/non-cold acclimation (Figure 6 and Appendix A). Transcriptome data showed that compared with cold acclimation, the expression of *HDT* genes was more strongly induced by cold stress and freezing stress after non-cold acclimation. Compared to other genes, the expression of *HDT2* gene pairs (*BnaC02g09720D* and *BnaA02g05840D*, *BnaC09g36350D* and *BnaA10g13800D*), the gene *BnaC04g39350D* of *HDT4* and the gene *BnaC01g23930D* of *HDT1* were particularly strongly expressed in response to cold stress and chilling stress. After cold acclimation of HX17 material, the expression of most genes induced by cold injury was stronger than that induced by freezing injury. After non-cold acclimation, the expression of most genes of HX17 material were all induced by cold and chilling injury. After cold acclimation of HX58 material, *BnaC02g09720D*, *BnaA02g05840D* and *BnaC09g36350D* of *HDT2* and *BnaC04g39350D* of *HDT4* were slightly induced by cold stress and freezing stress. After non-cold acclimation of HX58 material, the expression of *BnaC02g09720D*, *BnaA02g05840D*, *BnaA03g08590D*, *BnaC09g36350D* and *BnaA10g13800D* of *HDT2* as well as *BnaC04g39350D* of *HDT4* and *BnaC01g23930D* of *HDT1* were strongly induced by cold stress and chilling stress. In conclusion, some members of the *HDT* gene family are induced by cold and freezing stresses.

Colored rectangles indicate FPKM or TPM values; gray rectangles indicate no data. HX17/58_MA represents untreated leaves of 6-week-old HX17 or HX58 (two early-maturing semi-winter *B. napus* varieties) seedlings. CA represents leaves of 6-week-old seedlings treated with cold acclimation (4 °C for two weeks) and then treated at 4 °C for 12 h. FA represents leaves of 6-week-old seedlings treated with cold acclimation (4 °C for two weeks) and then treated at −4 °C for 12 h. MB represents untreated leaves of 6-week-old seedlings. CB represents leaves of 6-week-old seedlings treated at 4 °C for 12 h. FB represents leaves of 6-week-old seedlings treated at −4 °C for 12 h. The expression levels (log10(FPKM) value) of *HDT* genes are indicated by differently colored rectangles. Gene name color: red represents *BnaHDT1*, purple represents *BnaHDT2*, blue represents *BnaHDT3*, and brown represents *BnaHDT4*.

### 2.7. Expression Profiling of HDT Genes under Biotic Stress Stress

In addition, we found that *HDT* also responds to biological stress. The transcriptional profiling of *B. napus* susceptible (Westar) and tolerant (ZY821) genotypes infected with *S. sclerotiorum* (GSE81545: https://www.ncbi.nlm.nih.gov/geo/qu-ery/acc.cgi?acc=GSE81545, accessed on 24 May 2022) (Appendix A) showed that most of the *BnaHDT* genes were upregulated with *S. sclerotiorum*. In addition, the expression level was higher in tolerant plants than susceptible plants (Figure 7). The expression of *BnaC02g09720D* in susceptible plants (Westar) was higher than that in tolerant plants (ZY821). In summary, some members of the *HDT* family are induced by *S. sclerotiorum*.

Colored rectangles indicate FPKM or TPM values; gray rectangles indicate no data. Gene name color: red represents *BnaHDT1*, purple represents *BnaHDT2*, blue represents *BnaHDT3*, and brown represents *BnaHDT4*. 

Figure 7 is a heat map of *BnaHDT* expression levels (FPKM values) in susceptible (Westar) and tolerant (ZY821) genotypes of *B. napus* infected with *S. sclerotiorum*. Control: mock inoculated; 24 hpi; 24 h post-inoculation. The expression levels (log10(FPKM) value) of *HDT* genes are indicated by differently colored rectangles.

## 3. Discussion

The *HDT* family is a set of plant-specific histone deacetylases which plays a significance role in regulating plant development and resistance to biotic and abiotic stresses [19,22,28]. In *Arabidopsis*, *HDT* has four members, *HD2A* (*HDT1*), *HD2B* (*HDT2*) [14], *HD2C* (*HDT3*) [19] and *HD2D* (*HDT4*) [20,21]. In this study, we found 14, 10 and 8 HDT genes in *B. napus*, *B. rapa* and *B. oleracea*, respectively (Figure 1). In *B. napus*, the number of HDT genes in the An subgenome (7) and Cn subgenome (7) is almost the same as that in their diploid ancestors *B oleracea* (10) and *B. rapa* (8) (Table 1). This showed that most of the duplicated *HDT* genes were preserved after the whole genome duplication (WGD) event in *B. napus*. Homology analysis showed that three genes in *B. oleracea* (*Bo2g024200*, *Bo2g024210* and *Bo2g024220*) and one gene in *B. rapa* (*Bra009513*) were lost during or after the WGD event in *B. napus*. *HDT* sequence alignment (Appendix A) revealed that most *HDTs* are conserved in Brassicaceae, indicating that these duplicated *HDT* genes can still retain the function of their ancestors in these species. 

Among the *HDT* gene family, *HDT2* is expanded in *B. napus*. The immediate consequences are threefold: The first is the weakening of the function of the *HDT2* gene. The expression of *Arabidopsis HDT2* gene occurs in almost all tissues, while some *HDT2* genes are only specifically expressed in flowers, buds and ovule siliques in *B. rapa*, *B. oleracea* and *B. napus*, and some *HDT2* genes are not expressed in all tissues. These loss-of-function members may be short *HDT2* members (Figure 4). The second is tissue expression specificity. *BnaA02g05790D* and *BnaCnng34610D* are a pair of members with specifically high expression in ovules. A pair of genes (*BnaC02g09720D* and *BnaA02g05840D*) were expressed in all tissues. The final conseuence is the deletion of some *HDT2* genes. There are seven *HDT2* genes in *B. rapa* and seven *HDT2* genes in *B. oleracea*. Theoretically there should be 14 *HDT2* genes for *B. napus*, but in practice there are only 10, indicating that some *HDT2* genes are lost. These variations have a certain impact on the growth and development of rapeseed and help it to better adapt to the environment. Compared with *B. rapa* and *B. oleracea*, *BnaHDT2* is specifically expressed in flowers; presumably, *HDT2* has attained a new function in *B. napus*. 

*HDT* is regulated by various environmental stresses, has different tissue-specific expression patterns and has common and different functions in various developmental and physiological processes [14]. In *Arabidopsis*, *HDT* is mainly highly expressed in flowers, young siliques, stems, leaves and other tissues [14,19]. Like *Arabidopsis*, *HDT* genes of *B. napus*, *B. rapa* and *B. oleracea* are more highly expressed in reproductive organs such as flowers and young siliques. Yongfeng Hu et al. suggested that overexpression of *HDT701* induced earlier flowering under long-daylight conditions in hybrid rice, as shown in Figure 4, and most *BnaHDTs* were highly expressed in stamens in *B. napus* [24], suggesting that *BnaHDT* may also have a regulatory effect on flowering. Marie Lagac’e found that *ScHDT1* in potato (*Solanum chacoense*) increased the accumulation of *HDT* transcripts in ovules after fertilization [27], and we found that most *HDT2* genes were specifically expressed in flowers, buds and ovules of *B. napus*, *B. rapa* and *B. oleracea*. It is speculated that *HDT2* is related to reproductive growth. The expression of *Arabidopsis HDT4* in different tissues is lower than that of other *HDT* members. Similar to *Arabidopsis*, the expression of *HDT4* in various tissues of *B. napus*, *B. rapa* and *B. oleracea* was lower than that of other *HDT* members. The expression of *Arabidopsis HDT2* in different tissues was higher compared to other *HDT* members. As with *Arabidopsis*, *HDT2* (*BnaA02g05840D*, *BnaC02g09720D*) in *B. napus*, *B. rapa* and *B. oleracea* was also expressed more in various tissues than other *HDT* members. In summary, the function of *HDT* is conserved during gene evolution. Interestingly, we also found that most *HDT* genes are redundantly expressed in reproductive organs such as flowers and shoots, from which we can speculate that *HDT* genes have an important role in the reproductive functions of plants. This result indicates that *HDT* family genes have conserved and specific biological functions in various developmental and physiological processes of *B. napus*. 

It is well known that ABA is an indispensable hormone in plant stress response and plays an important role in abiotic stress. ABA inhibited the expression of *AtHDT3* in *Arabidopsis* [18]. We found that most *BnaHDTs* were induced by ABA (Figure 5). In *Arabidopsis*, *HDT3* is responsive to cold stress, whereas in our study, *BnaHDT2* expression was found to be strongly induced by cold induction for reasons that remain to be explored subsequently. Furthermore, most members of *HDT* of *B. napus*, especially *BnaC02g09720D* and *BnaA02g05840D*, were slightly induced by low-temperature stress but were strongly induced by low temperature (4 °C) and freezing (−4 °C) under non-cold acclimation conditions. *BnaHDT3* specifically responds to heat stress, is barely expressed under cold and freezing stress and is induced after infection with S. sclerotiorum in tolerant (ZY821) genotypes of *B. napus*, suggesting that this gene may have a function in responding to biotic stresses. In conclusion, *HDT* genes can play an important role in resisting many adversity stresses, like heat, cold, freezing, ABA and infection of *S. sclerotiorum* in *B. napus*. This suggests a functional differentiation of the *HDT* family.

## 4. Materials and Methods

### 4.1. Identification of the HDT Gene Family

TAIR (https://www.Arabidopsis.org/index.jsp, accessed on 13 August 2021) was used to obtain the nucleotide and protein sequences for *Arabidopsis AtHDT*. Using BLASTp (E-value < 1 × 10^−5^) in Ensembl genomes (http://ensemblgenomes.org/, accessed on 13 August 2021), four *AtHDT* proteins were used as query sequences to search for the HDT proteins of *B. napus*, *B. rapa* and *B. oleracea*. In ExPasy and Ensembl Plants (https://web.ex-pasy.org/compute_pi/, accessed on 13 August 2021), the molecular weights (MW), isoelectric points (IP) and subcellular localizations of HDT proteins were predicted [36].

### 4.2. Analysis of Gene Structure, Motif Composition

NCBI (https://www.ncbi.nlm.nih.g-ov/cdd) and Pfam (http://pfam.xfam.org, accessed on 16 August 2021) were used to characterize the HDT domain [37]. MEME (http://me-me.nbcr.net/meme/cgi-bin/meme.cgi, accessed on 27 August 2021) was used to analyze the conserved motifs. To examine the gene structures and motif composition, TBtools version 1.095 was used [38].

### 4.3. Phylogenetic Analysis and Chromosomal Locations

ClustalW was used to align the multiple sequences of all detected HDT proteins (from *Arabidopsis*, *B. napus*, *B. rapa* and *B. oleracea*), and MEGA X was used to build a phylogenetic tree using the neighbor-joining (NJ) phylogenetic technique with 1000 bootstrap replicates [39]. TBtools version 1.095 was used to examine the gene chromosomal localization [38].

### 4.4. Plant Materials and Treatments, Heat Map Analysis of the HDT Transcriptome Data

Rapeseed ZS11 (the semi-winter cultivar Zhongshuang 11) [33] seeds were germinated on filter paper, and the seedlings were then transplanted into pots with soil or vermiculite and nurtured in a growth chamber for six weeks. Based on RNA-Seq data, the expression patterns of ZS11’s roots, stems, leaves, buds, sepals, stamens, new pistils, blossoming pistils, wilting pistils, siliques, pericarps and ovules were examined [40]. *B. rapa*, and *B. oleracea*’s RNA-Seq data were evaluated to determine the pattern of expression in various tissues [41].

Hormone treatments were performed by sprinkling leaves with 100 μM abscisic acid (ABA), 100 μM methyl jasmonate (MeJA), 1 mM salicylic acid (SA) and 10 μg/mL ethephon (ETH) solutions during the 8:00 a.m.–8:00 p.m. period. To simulate salt and PEG stresses, seedlings were righted with NaCl (200 mM) or PEG-6000 (20%) solution. To simulate hot and cold stresses, seedlings were grown in a chamber at 40 °C or 4 °C. Leaf samples were collected at 3, 6 and 12 h time points during stress treatment. For chilling and freezing treatments with or without cold acclimation, two early-maturing semi-winter rapeseed varieties (HX17 and HX58) were used [42]. They were treated as described previously. Seedlings were cultured in incubators under 20 °C (14 h light: 6:00 a.m.–8:00 p.m.)/16 °C (10 h dark: 8:00 p.m.–6:00 a.m.) for 4 weeks and then treated at 4 °C (14 days)→4 °C (12 h) (CA) or −4 °C (12 h) (FA), 20 °C/16 °C (light/dark) for 6 weeks→4 °C (12 h) (CB), 20 °C (14 h light: 6:00 a.m.–8:00 p.m.)/16 °C(10 h dark: 8:00 p.m.–6:00 a.m.) for 6 weeks→−4 °C (12 h) (FB). For the acclimation condition, after the 14 days at 4 °C, 4 °C/−4 °C (12 h), they were treated at 4 °C or −4 °C from 8:00 p.m.–8:00 a.m. (10 h dark and 2 h light) [43]. Then, the third leaves from the top were collected at 8:00 a.m. after cold treatment and stored at −80 °C immediately until RNA extraction. 

Expression data after inoculation with *S. sclerotiorum* were obtained from the GEO database (GEO: GSE81545) [44]. TBtools version 1.095 was employed to generate heat maps of the expression profile values of *BnaHDT* genes [38].

## 5. Conclusions

In this study, we identified 14, 8 and 10 HDT proteins in *B. napus*, *B. rapa* and *B. oleracea*, respectively, by exploring the important role of *HDT* genes in *Brassicaceae* plants. It was found that the *HDT2* homologous gene of *Brassicaceae* plants was super-amplified, and all HDT proteins had conserved NPL (nucleoplastin-like) domains. In addition, we found that the *Bna/Br/BoHDT* gene was expressed differently in different tissues at different developmental stages, mainly in flowers and pods, and responded to low-temperature and freezing stresses. This study will lay a foundation for further understanding the biological function of *HDT* subfamily genes in *B. napus* and the study of low-temperature stress, especially freezing stress.

## Figures and Tables

**Figure 1 ijms-24-00525-f001:**
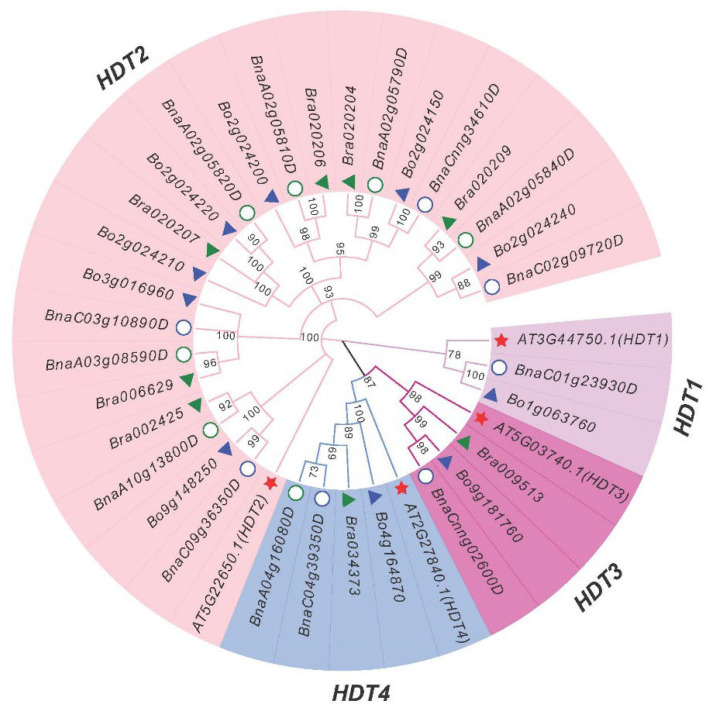
Phylogenetic analysis of 36 HDT proteins from *B. napus* (14), *B. rapa*, (8), *B. oleracea* (10) and *Arabidopsis* (4).

**Figure 2 ijms-24-00525-f002:**
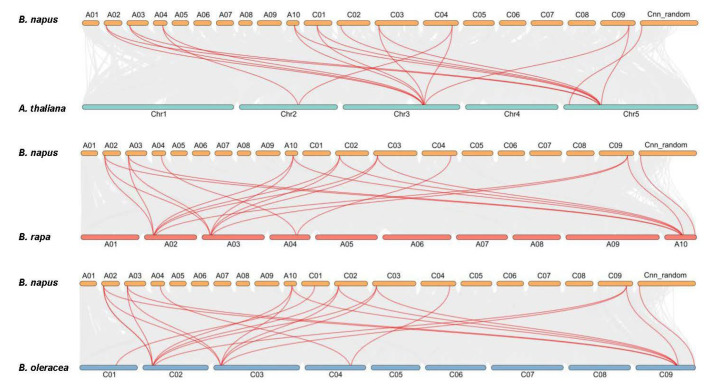
Syntenic relationship of *HDAC* genes in *B. napus* and three ancestral plant species. Grey lines in the background show the collinear blocks within rapeseed and other plant genomes, while the red lines highlight the syntenic *HDAC* gene pairs.

**Figure 3 ijms-24-00525-f003:**
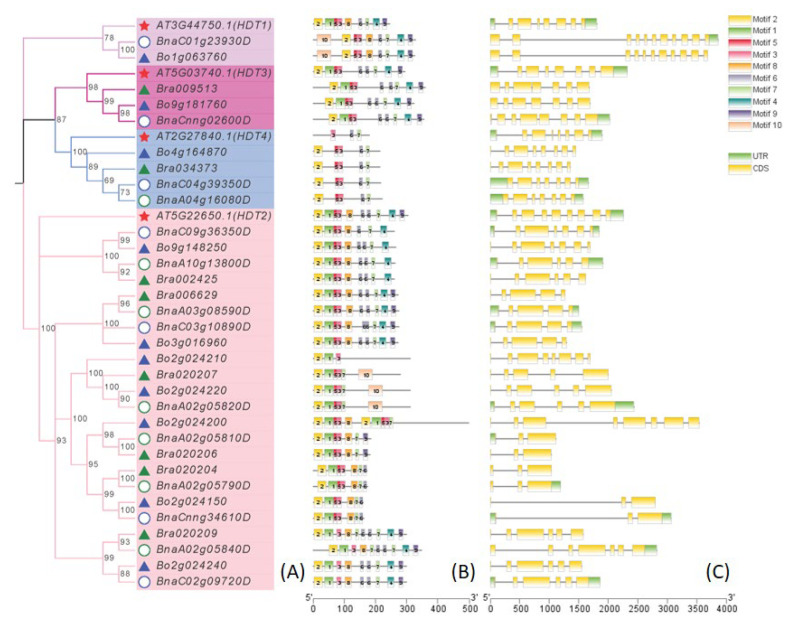
Phylogenetic tree (**A**), gene motifs (**B**) and gene structure (**C**) of *HDT* of *Arabidopsis*, *B. rapa*, *B. oleracea* and *B. napus*. Neighbor-joining phylogenetic tree showing the relationship among 8 *B. rapa* (green triangle), 10 *B. oleracea* (blue triangle), 14 *B. napus* (circle) and 4 *Arabidopsis* HDT proteins (**A**). The resulting four groups are labeled (Group I–IV). Ten motifs in HDT proteins were identified by MEME tools (**B**). Yellow boxes, black lines and green boxes indicate exons, introns and untranslated regions, respectively (**C**).

**Figure 4 ijms-24-00525-f004:**
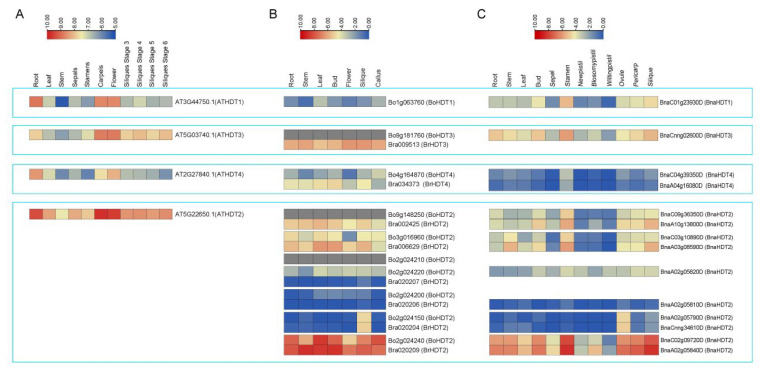
Expression of *AtHDT* (**A**), *BraHDT* and *BoHDT* (**B**), *BnaHDT* (**C**) in different tissues and organs. The expression levels (log10(FPKM) value) of *HDT* genes were indicated by differently colored rectangles.

**Figure 5 ijms-24-00525-f005:**
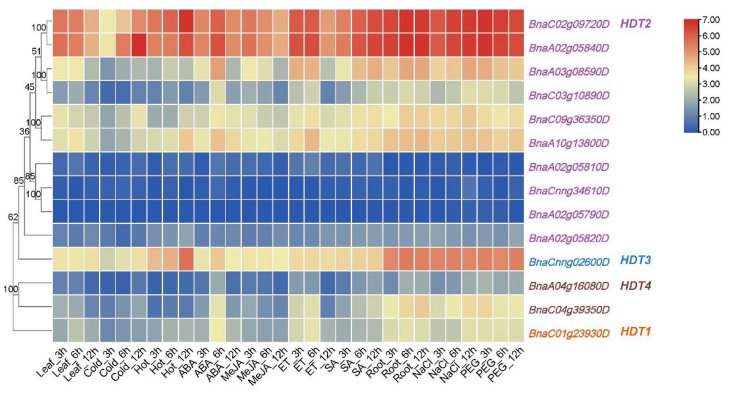
Expression of *BnaHDT* genes under different abiotic stresses and plant hormone treatments. The expression levels (TPM values) of *BnaHDT* genes are indicated by differently colored rectangles. Leaf: untreated leaves; Cold: leaves treated with −4 °C. Hot: leaves treated with 40°C; ABA: leaves treated with 100 μM abscisic acid; MeJA: leaves treated with 100 μM methyl jasmonate; ETH: leaves treated with 10 μg/mL ethephon; SA: leaves treated with 1.0 mM salicylic acid. Root: untreated roots; NaCl: roots treated with 200 mM NaCl; PEG: roots treated with 20% polyethylene glycol 6000. The expression levels (log10(FPKM) value) of *HDT* genes are indicated by differently colored rectangles. Gene name color: Red represents *BnaHDT1*, purple represents *BnaHDT2*, blue represents *BnaHDT3*, and brown represents *BnaHDT4*.

**Figure 6 ijms-24-00525-f006:**
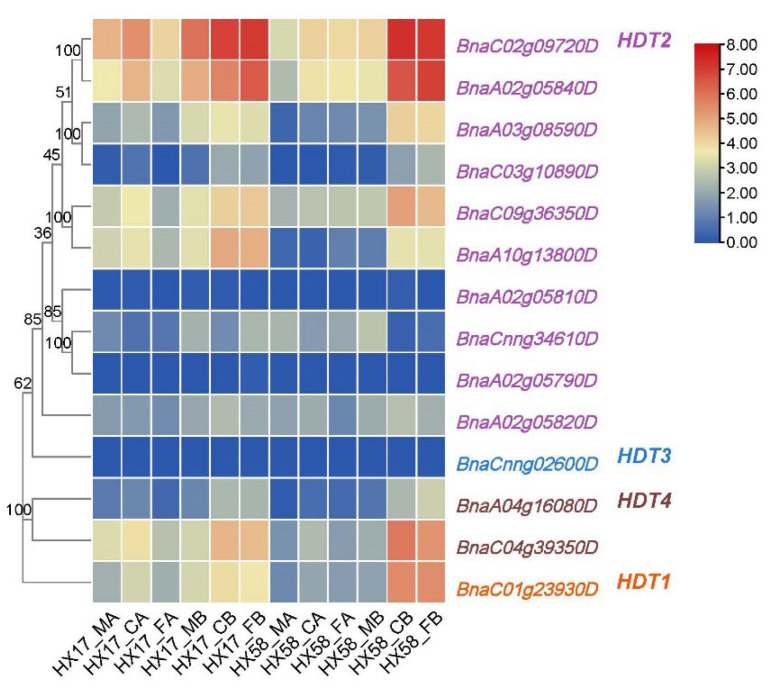
Expression of *BnaHDT* genes under different low-temperature treatments.

**Figure 7 ijms-24-00525-f007:**
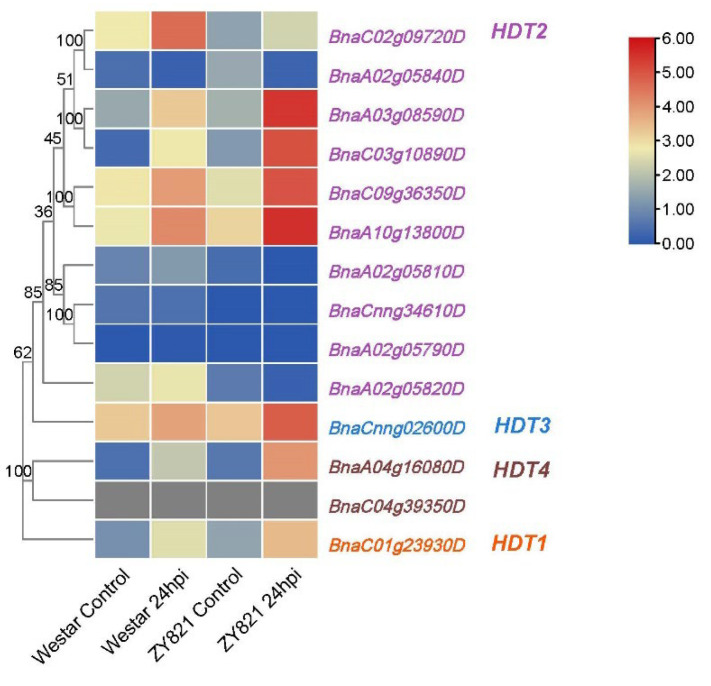
Expression of *BnaHDT* genes under infection with *S. sclerotiorum* (B).

**Table 1 ijms-24-00525-t001:** List of *HDT* genes identified in *Arabidopsis*, *B. rape*, *B. oleracea* and *B. napus*.

	Homologous Gene in *B. rape/B. oleracea/B. napus*	Gene ID	Nucleotide Length (bp)	Amino Acid	Domains	Isoelectric Point Prediction	Molecular Weight (kD)	Number of Introns	Number of Exons	Predicted Subcellular Localization	Chromosome Location
** *HDT1* **	** *Arabidopsis* **	*AT3G44750*	738	245	NPL	4.8566	26.372	7	8	Nuclear	Chr3:16,297,656-16,299,876
	** *B. rape* **	*NO*									
	** *B. oleracea* **	*Bo1g063760*	984	327	NPL	4.8506	35.48	10	11	Nuclear	C9:53771260-53773620
	** *B. napus* **	*BnaC01g23930D*	984	327	NPL	4.8506	35.48	10	11	Nuclear	ChrC01:371,077-374,932
** *HDT2* **	** *Arabidopsis* **	*AT5G22650*	918	305	NPL	4.4405	32.218	8	9	Nuclear	Chr5:7,534,017-7,536,273
	** *B. rape* **	*Bra002425*	786	261	NPL	4.391	28.085	5	6	Nuclear	A10:9,653,453-9,655,071
	** *B. oleracea* **	*Bo9g148250*	801	266	NPL	4.6089	28.727	6	7	Nuclear	C9:43880123-43881813
	** *B. napus* **	*BnaC09g36350D*	783	260	NPL	4.7321	28.027	6	7	Nuclear	ChrC09:229,647-231,492
		*BnaA10g13800D*	786	261	NPL	4.3778	27.973	5	6	Nuclear	ChrA10:60,660-62,567
	** *B. rape* **	*Bra006629*	822	273	NPL	4.3589	29.371	4	5	Nuclear	A03:4,353,992-4,355,262
	** *B. oleracea* **	*Bo3g016960*	819	272	NPL	4.418	29.167	4	5	Nuclear	C3:5486467-5487752
	** *B. napus* **	*BnaA03g08590D*	828	275	NPL	4.3442	29.495	4	5	Nuclear	ChrA03:952,360-953,862
		*BnaC03g10890D*	828	275	NPL	4.4263	29.574	4	5	Nuclear	ChrC03:3,176,327-3,177,870
	** *B. rape* **	*Bra020207*	840	279	NPL	8.8665	31.737	4	5	NONE	ChrA02:5,568,145-5,570,144
	** *B. oleracea* **	*Bo2g024210*	939	312	NPL	6.96	35.64	7	8	NONE	C2:6606801-6609175
		*Bo2g024220*	939	312	NPL	8.0531	35.565	5	6	NONE	ChrC2:6614008-6616054
	** *B. napus* **	*BnaA02g05820D*	933	310	NPL	8.3296	35.276	5	6	NONE	ChrA02:390,454-392,883
		*NO*									
	** *B. rape* **	*Bra020206*	555	184	NPL	4.3025	19.974	2	3	Nuclear	A02:5,565,814-5,566,844
	** *B. oleracea* **	*Bo2g024200*	1494	497	NPL	6.2865	56.102	7	8	NONE	C2:6598993-6603942
	** *B. napus* **	*BnaA02g05810D*	555	184	NPL	4.3025	19.974	2	3	Nuclear	ChrA02:387,265-388,818
		*NO*									
	** *B. rape* **	*Bra020204*	525	174	NPL	3.9815	18.988	2	3	Nuclear	A02:5,555,265-5,556,304
	** *B. oleracea* **	*Bo2g024150*	489	162	NPL	3.881	17.763	2	3	Nuclear	C2:6578131-6580929
	** *B. napus* **	*BnaA02g05790D*	525	174	NPL	3.9815	18.988	2	3	Nuclear	031993:378,511-379,705
		*BnaCnng34610D*	489	162	NPL	3.881	17.763	2	3	Nuclear	ChrCnng:33,627-36,691
	** *B. rape* **	*Bra020209*	903	300	NPL	4.4385	31.901	5	6	Nuclear	A02:5,572,571-5,574,144
	** *B. oleracea* **	*Bo2g024240*	903	300	NPL	4.4385	32.051	5	6	Nuclear	C2:6623475-6625018
	** *B. napus* **	*BnaA02g05840D*	1047	348	NPL	4.4926	36.905	6	7	Nuclear	ChrA02:393,948-396,765
		*BnaC02g09720D*	903	300	NPL	4.4192	31.939	5	6	Nuclear	ChrC02:292,978-294,839
** *HDT3* **	** *Arabidopsis* **	*AT5G03740*	885	294	NPL	4.5108	31.830	7	8	Nuclear	Chr5:981,859-984,289
	** *B. rape* **	*Bra009513*	1083	360	NPL	4.468	39.679	6	7	Nuclear	A10:17,010,728-17,012,399
	** *B. oleracea* **	*Bo9g181760*	972	323	NPL	4.3532	35.343	6	7	Nuclear	C9:53,771,597-53,773,283
	** *B. napus* **	*NO*									
		*BnaCnng02600D*	1071	356	NPL	4.5045	39.242	7	8	Nuclear	ChrCnng:25,960-27,981
** *HDT4* **	** *Arabidopsis* **	*AT2G27840*	546	181	NPL	3.9932	20.073	6	7	Nuclear	Chr2:11,861,806-11,863,908
	** *B. rape* **	*Bra034373*	648	215	NPL	4.387	23.75	7	8	Nuclear	A04:12,193,709-12,195,064
	** *B. oleracea* **	*Bo4g164870*	648	215	NPL	4.4398	23.793	7	8	Nuclear	C4:44092147-44093599
	** *B. napus* **	*BnaA04g16080D*	663	220	NPL	4.4965	24.205	6	7	Nuclear	ChrA04:234,940-236,511
		*BnaC04g39350D*	648	215	NPL	4.387	23.739	6	7	Nuclear	ChrC04:1,402,081-1,403,742

## Data Availability

All relevant data are available from the corresponding author on request (xiepan@hunau.edu.cn).

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
