# Peer review of "Comprehensive Analyses of the Histone Deacetylases Tuin (HDT) Gene Family in Brassicaceae Reveals Their Roles in Stress Response"

_ijms, 2022, doi:10.3390/ijms24010525_

Round 1
Reviewer 1 Report
The manuscript reports on the analysis of Histone Deacetylase Tuin (HDT) genes in brassica species including phylogeny and gene expression.
Overall the paper was well presented and contains some interesting data about this gene family. The discussion was a fair interpretation of the results obtained.
Minor comment:
line 199: sentence is incomplete
Author Response
Point 1: line 199: sentence is incomplete
Response 1: Thank you for your valuable sugession. We have completed the sentences as following: To reveal roles of BnaHDTs on stress-responsiveness in B. napus, their expression pattern upon various abiotic and phytohormone treatments were investigated.

Reviewer 2 Report
The manuscript is clear and well written. Authors explain clearly the aim of this work. The experiments are well conducted and the figures are clear. However some details should be revised.
In the Introduction (lines 48-54) lacks important recent reference in this field concerning HDAC family in plants and Brassicaceae (Trends in Plant Science, 2021, DOI:https://doi.org/10.1016/j.tplants.2020.12.011 ). Authors should be updated the investigations and new results, and discuss the phylogeny and new functionalities of HDAC superfamily in Brassicaceae to better understanding their functions and evolution.
In Materials and Methods, authors use ClustalW for sequence alignment before phylogeny. They should be use a more robust alignment such as ClustalOmega.
In Figure 1 lacks statistical analysis. The bootstrap values should be shown in the branches of phylogenetic tree. Also mention in Materials and Methods.
In Discussion, to better understanding the role of plant HDT family in the context of HDAC superfamily, authors could discuss the redundancy or not redundancy of their functions and their putative importance.
I consider that this paper will be of interest for researchers working in plants and particularly in the field of HDAC and can be published with revision.
Additional comments:
The taxonomic name of all plants should be written in italics along the text ( i.e. B. napus, B. rapa, and B. oleracea). Write the complete name of plant the first time it is cited, and after use the abbreviation. For instance, in line 281, B. napus should be written instead Brassica napus.
Author Response
Point 1: In the Introduction (lines 48-54) lacks important recent reference in this field concerning HDAC family in plants and Brassicaceae (Trends in Plant Science,2021, DOI:https://doi.org /10.1016/j.tplants.2020.12.011). Authors should be updated the investigations and new results, and discuss the and new functionalities of HDAC superfamily in Brassicaceae to better understanding their functions and evolution.
Response 1: Thank you for referring us to the valuable literature, which we have cited in the paper.
Point 2: In Materials and Methods, authors use ClustalW for sequence alignment before phylogeny. They should be use a more robust alignment such as ClustalOmega.
Response: Thank you for your valuable sugession. We used Clustal Omega, ClustalW and Muscle
alignment to analyze all HDT protein sequences ( S1). Based on the blast and the previous results of conserved domains and motifs (Figure 3), we found that the cluster analysis of ClustalW was consistent with the previous results, so we infer that the phylogenetic tree of ClustalW is the best method corresponding to the previous results.
Point 3: In Figure 1 lacks statistical analysis. The bootstrap values should be shown in the branches of phylogenetic tree. Also mention in Materials and Methods.
Response: Thank you for your important sugession. We have replaced Figure 1 with bootstrap values with Figure 1 without bootstrap values.
Point 4: In Discussion, to better understanding the role of plant HDT family in the context of HDAC superfamily, authors could discuss the redundancy or not redundancy of their functions and their putative importance.
Response: Thank you for your valuable sugession. We have added the following description:In summary, the function of HDT is conserved during gene evolution. Interestingly, we also found that most HDT genes are redundantly expressed in reproductive organs such as flowers and shoots, from which we can speculate that HDT genes have an important role in the reproductive function of plant.
Point 5: The taxonomic name of all plants should be written in italics along the text ( i.e. B. napus, B. rapa, and B. oleracea). Write the complete name of plant the first time it is cited, and after use the abbreviation. For instance, in line 281, B. napus should be written instead Brassica napus.
Response: Thank you for reminding us the improper format on the study.The irregularities in the format of the full text have been corrected, and all gene names, taxonomic names, and genus names of all plants have been changed to italics.
